# Computer-Aided Evaluation of Interstitial Lung Diseases

**DOI:** 10.3390/diagnostics15070943

**Published:** 2025-04-07

**Authors:** Davide Colombi, Maurizio Marvisi, Sara Ramponi, Laura Balzarini, Chiara Mancini, Gianluca Milanese, Mario Silva, Nicola Sverzellati, Mario Uccelli, Francesco Ferrozzi

**Affiliations:** 1Department of Radiology, Istituto Figlie di San Camillo, 26100 Cremona, Italy; 2Department of Internal Medicine and Pneumology, Istituto Figlie di San Camillo, 26100 Cremona, Italy; 3Scienze Radiologiche, Dipartimento di Medicina e Chirurgia, University Hospital of Parma, 43126 Parma, Italy

**Keywords:** interstitial lung disease, AI (artificial intelligence), hierarchical learning

## Abstract

The approach for the diagnosis and treatment of interstitial lung diseases (ILDs) has changed in recent years, mainly for the identification of new entities, such as interstitial lung abnormalities (ILAs) and progressive pulmonary fibrosis (PPF). Clinicians and radiologists are facing new challenges for the screening, diagnosis, prognosis, and follow-up of ILDs. The detection and classification of ILAs or the identification of fibrosis progression at high-resolution computed tomography (HRCT) is difficult, with high inter-reader variability, particularly for non-expert radiologists. In the last few years, various software has been developed for ILD evaluation at HRCT, with excellent results, equal to or more reliable than humans. AI tools can classify ILDs, quantify the extent, analyze the features hidden from the human eye, predict prognosis, and evaluate the progression of the disease. More advanced tools can incorporate clinical and radiological data to obtain personalized prognosis, with the potential ability to steer treatment decisions. To step forward and implement in daily practice such tools, more collaboration is required to collect more homogeneous clinical and radiological data; furthermore, more robust, prospective trials, with the new AI-derived biomarkers compared with each other, are needed to demonstrate the real reliability of the computer-aided evaluation of ILDs.

## 1. Introduction

During the last two decades, artificial intelligence (AI) has reached crescent popularity in medicine, particularly in radiology. Even though AI technology was developed in the 1940s, in only the last few years it has been seriously applied in medicine. The diagnosis of interstitial lung diseases (ILDs) is based on clinical, functional, radiological, and pathological data [1]. Following the last guidelines provided by the American Thoracic Society (ATS)\European Respiratory Society (ERS)\Japanese Respiratory Society (JRS)\Latin American Thoracic Society (ALAT), the diagnosis of ILDs is mainly based on a multidisciplinary discussion (MDD), with great importance of high-resolution computed tomography (HRCT) features [1,2]. When the MDD was firstly introduced, the main goal of the ILD evaluation was to differentiate patients affected by idiopathic pulmonary fibrosis (IPF) from other ILDs, since IPF carries a progressive behavior and can be treated with antifibrotic drugs, such as pirfenidone or nintedanib [1]. Recently, it has been demonstrated that several ILDs other than IPF can be progressive, and they were defined as progressive pulmonary fibrosis (PPF) [3]. Similarly to IPF, antifibrotic therapy can reduce the PPF progression [4]. Nevertheless, the role of the radiologist within the MDD is difficult, mainly for two reasons: (1) the evaluation of the HRCT pattern is affected by high inter-reader variability; (2) the detection of little modifications in disease extent or features during short follow-up has low sensitivity [5]. AI techniques can potentially overcome these two limitations, helping radiologists to identify the correct ILD pattern and to detect minimal HRCT modifications over time.

The aim of this review is to describe the application of AI that could aid clinicians and radiologists to solve the challenges in the diagnosis and follow-up of ILDs.

## 2. Computer-Aided Techniques

Before describing computer-aided techniques, it is necessary to underline some specific features of medical imaging. As compared with photographic images, medical imaging has low spatial resolution but higher contrast resolution. Typically, MR images have a size of 256 × 256, while CT images have a size of 512 × 512. The grayscale information of a CT image is 12 bits, which is higher than perceived by the human eye, thus multiple contrast and brightness settings should be used to detect all information. This is translated into window width and center. In CT, all pixels have an intensity value, which is 0 for water and −1000 for air, using the Hounsfield Unit (HU) scale. The intensity that can be perceived by human eye ranges from center-width\2 to center+width\2. Nevertheless, not all images (for example magnetic resonance or ultrasound) have an intensity value associated, thus a normalization process of the images is usually required. For this reason, in the 1980s, a standard system for storing medical images was developed, called Digital Imaging and Communications in Medicine (DICOM). DICOM images have a header and a body. The header is composed of keys (set of standard codes and tags) and values (encoded in a prescribed way). The body instead is constituted by pixel values.

The computer-aided techniques employed in medical imaging require preparation of the images. Nowadays medical imaging datasets consist of thousands of images per patient stored in several gigabytes. These datasets are not structured, and preprocessing is needed before the beginning of imaging analysis. During preprocessing, informative characteristics are enhanced, and simultaneously images noise is reduced. Preprocessing steps are contrast adjustment, denoising, edge enhancement, registration, and advanced transformation. After preprocessing, the sampling process begins and can be conducted on pixels, arbitrary regions, segmented organs, or the entire image. In ILDs, entire lungs or regions of interest with lung modifications were previously segmented manually or semi-automatically. Then, features were extracted from images by mathematical operations that transform images in number descriptors of texture, shape, and other characteristics that can be associated with clinical parameters. The more common feature extraction are transformations, first- and second-order statistics, and filter-based methods [6,7,8].

After the preparation of the images, the three main computer-aided techniques applied in ILDs were quantitative computed tomography (QCT), machine learning, and deep learning. QCT analysis was the first method applied in ILDs to overcome the poor reproducibility of visual assessment. One of the QCT applications was lung densitometry, which calculates the number of lung pixels included in a specific range of attenuation values (HU). Primarily, this technique was employed to estimate the volume of emphysema, defined as the volume of the lung under −950 HU [9]. A similar approach was proposed for the quantification of ILDs. The pixels included between −250 and −600 HU, defined as high attenuation areas (HAAs), correlated with pulmonary function tests (PFTs) and transplant-free survival in patients with biopsy-proven IPF [10]. Another QCT application is based on the analysis of lung histogram. Several features can be extracted from the lung histogram, such as percentiles and first-order statistics, including mean, median, skewness, and kurtosis. For the emphysema quantification, the estimation of the lung with attenuation values under the 15th percentile of the density histogram was proposed [11]. The use of the 15th percentile of the density histogram is more robust for detecting emphysema changes over time and less affected by changes in lung volume in comparison with the use of a standardized attenuation value threshold [12]. Similarly, in IPF patients, an increase over time at HRCT in the 40th percentile of the density histogram can reflect fibrosis progression, while an increase in the 80th percentile identifies augmented reticulations (Figure 1) [13]. The skewness of the density histogram measures the asymmetry of the attenuation value distribution, with negative skewness indicating longer left tail and more pixels toward high attenuation values. The kurtosis indicates the tailedness of the density histogram, with low kurtosis reflecting a reduced number of pixels with normal ventilated lungs. As expected, decreased skewness and kurtosis are correlated with reduced disease progression and mortality in patients with IPF [14]. QCT measurements are currently easy to obtain; nevertheless, they carry several limitations in the quantification of ILDs: (1) scarce differentiation of variegate patterns that constitute ILD, composed by alterations with opposite attenuations (reticulations, consolidations, and ground-glass opacities with high attenuation vs. honeycombing with low attenuation); (2) no localization of lung alterations (distribution of alterations on axial or sagittal planes); (3) influence of CT acquisition parameters (different inspiration with careful patient coaching by the technologist required and use of smooth reconstruction algorithm) [15].

Machine learning is a kind of AI in which computers do a given task based on sample data (Figure 2). Features are the starting point for the machine learning methods. Features are numerical values associated with an image. When several features for one example are mixed, they become a vector. For medical images, features could be all numerical values that can be computed from pixels. There are also non-medical image features, such as patient age or gender and laboratory test results that can be combined with values derived from medical images, forming a vector. If more data are available for an example, and more examples are included in the datasets, the machine learning process could be more precise. Nevertheless, several features are not informative and may overlap with other features and should be deleted. This step is called feature selection or reduction. The selection of the informative features reduce the time required to accomplish the task by the machine learning model. To reduce the number of features, three methods are used: (1) the filter-based method, which relies on Pearson correlation and chi-square to determine if a feature is predictive and independent from other features; (2) the wrapper method, which removes features with the aim to detect the features that reduce minimally the performance of the model; and (3) embedded methods, which are so-called considering that they are built into the machine learning process (the more common are lasso and random forest). To accomplish the task, machine learning can analyze data with different models. The easiest is logistic regression, which outputs a binary category prediction based on several input features (for example, if a person will have or not have a heart attack on the basis of weight, age, etc.). Logistic regression is based on a sigmoid function from 0 to 1; when the output is less than 0.5, the example is assigned to one class, if higher than 0.5, to the other class. Logistic regression is easy to implement in machine learning methods and provides valuable insights; nevertheless, it should be applied only when several key assumptions are satisfied: (1) binary output; (2) no collinearity between input features; (3) a large sample size in relation to the number of input features; (4) minimum outliers; and (5) independent observations between input features. Another model employed is the *decision trees* or *random forests*. Decision trees make a series of binary decisions until a final decision is made. A range of values are tried, and the best threshold is the one that makes the majority of the cases right. In contrast with logistic regression, by using decision trees, it is possible to obtain both categorical and continuous output. During the process the features and the threshold of the features are identified. The Gini index or entropy (or information gain) are used for detecting categorical features, while mean error or regression are used for continuous variables. After features are selected, the best threshold for each feature is selected. This process is repeated several times until a stopping criterion is met (for example, a decided number of decisions is made). Decision trees are easily interpreted and graphically intuitive to understand how the model works. Support vector machines (SVMs) are algorithms with the aim of identifying a “hyperplane” that can separate values from one category to another. These algorithms can be used for linearly separable categories. The aim of SVMs is to identify a formula that can calculate the hyperplane that best separates the features into two categories. SVMs are popular and can be used for categorical and continuous variables. Indeed, binary categories can be put together to form polynomial SVMs, and continuous variables can be transformed in higher dimensions and transformed into categorical variables (the so-called “kernel trick”). Logistic regression, decision trees, and SVMs are supervised methods that have external labeled data with known input–output pairs. Unsupervised modeling is used to identify patterns and useful features from unlabeled data, without predefined outputs. Neural networks can be used for unsupervised tasks. Each input feature or vector is assigned to a neuron, defined node. Each node processes the vector, and then passes the information to another node at the next level with different weights that can strengthen or reduce the signal. Usually each node processes the information, and if it is below a threshold, it switches off the node of the next layer; by contrast, if the analysis is higher than a threshold, it switches on the node of the next layer. The analysis continues until an output layer is obtained.

Deep learning is a subset of machine learning that uses neural networks. Among deep learning methods, convolutional neural networks (CNNs) are considered translation-invariant, which means that they can recognize the pattern learned in any location or orientation. CNNs are trained with medical images and other patients’ data to identify relationships between features extracted and the class labels that constitute diagnosis. CNNs include four computer vision tasks, such as image classification, object detection, semantic segmentation, and instance segmentation (Figure 3). In image classification, CNNs predict the class of an entire image (e.g., chest diseased or normal). Object detection identifies a specific entity on an image (e.g., a liver metastasis). Semantic segmentation associates each pixel of an image with a specific class (e.g., blood in contrast-enhanced CT). Instance segmentation requires object detection and then assigns the object to a class (e.g., lung nodules at HRCT). During the training of a deep learning model, it is crucial to have labeled images representative of the task. Images from a single medical center are usually insufficient, while multicentric datasets are more representative but introduce biases related to non-standardized images and privacy issues. When datasets are constituted by a low number of patients, the model works perfectly on the training dataset but has poor performance on new data. To solve this issue, several techniques of data augmentation have been developed, such as random translations, rotation, flip, scaling, crop, and brightness or contrast adjustments; in addition, new techniques such as generative adversarial networks (GANs) are interesting methods that create fake images to expand training datasets (Figure 4). Another limitation of model training is data labeling. Labeling images is time-consuming and is usually based on reporting or expert review. A recent advance in this field is pseudo-labeling, which is a semisupervised method where a partially trained model predicts labels (pseudo-labels) which are included in the training dataset. CNNs are constituted by input values connected with hidden layers that finally give an output about a specific task, for example, recognizing an imaging pattern (Figure 5). The first layer is a convolution layer in which a pixel’s kernel is applied over all images. After that, a pooling procedure could be applied using a reduced kernel and the maximum value of each feature extracted (negative values are transformed to 0), obtaining a reduced size of the image. The convolution and the pooling process could be performed for several hidden layers; the first layers recognize the simple features of the image, such as lines and edges, while the last layers recognize the more complex components of the image. Then, an output layer is obtained, in which there are several one-dimensional vectors combined in a conventional neural network. If there are five output classes, and only one is correct, the right class should have 1 value and the others 0 value. Two critical elements characterize CNNs: weights and normalization. Weights are calculated by a back-propagation process. Before training CNNs on a specific task, the output is usually wrong. This error is defined as the loss of function. The gradient of loss is then used as a weight to correct the CNN algorithm error. Furthermore, the image requires normalization; for example, an image that is too dark or too bright can confuse the algorithm. Usually, the mean value is considered 0 and 1 value as a magnitude of 1. Nevertheless, with such an approach, important information can be lost.

After the training process, the model should be validated on an unseen dataset. Usually, the dataset is split into training, validation, and test set. Ideally, the real performance of the model should be tested on complete external datasets, with new patients or different geographical sites. The quantitative assessment of model performance is fundamental and must be appropriate for the task. For binary classifications, standard biostatistical metrics can be used by calculating accuracy, sensitivity, specificity, and positive and negative predictive values. The receiver operating characteristic curve (ROC) plots the true-positive rate versus the false-positive rate; the area under the curve (AUC) calculates the model performance, with 1 as perfect model and 0.5 as random model. When the task is multiclass detection, contingency tables called confusion matrices are useful to evaluate if a particular class tends to be confused. For object detection or segmentation, metrics are used that calculate how well the segmented area of an object matches the real segmentation, usually performed by a radiologist. The intersection is the area that overlaps the human segmentation, while the union is the total area of predicted and human segmented area. The intersection over union (IOU) and DICE scores are used for this evaluation, with a value of one as a perfect match (Figure 6). Visualization maps are useful for evaluating the areas within an image that change the model classification. In this regard, gradient-weighted class activation maps (Grad-CAMs) result in a heat map of important features that the model takes into account for the output probability or classification.

## 3. Challenges in ILD Evaluation

The first application of AI tools in chest imaging was lung nodules. The requirement of automated tools for the detection of lung nodules emerged with the development of screening programs for detection of lung cancer, with the recruitment of thousands of subjects. Deep learning methods demonstrated their ability to help radiologists in detecting lung nodules at low-dose thoracic CT [18]. Diagnosis and follow-up of ILDs is a more complex task. ILDs are constituted by different patterns with different locations and are a combination of fibrosis and inflammation with various appearances [19]. In addition, the etiology can vary from neoplastic to infectious or autoimmune disease, with a non-negligible percentage of idiopathic conditions. Deep learning can overcome three main challenges in ILDs: (1) early detection; (2) diagnosis and prognosis; and (3) monitoring of the disease, with particular interest in therapy effectiveness.

A condition that has recently received increasing interest is subclinical lung interstitial lung abnormalities (ILAs) [20]. ILAs are defined as incidental findings in a non-dependent zone affecting more than 5% of any lung zone, detected at partial or complete chest CT performed without suspicion of interstitial lung disease [20]. ILAs include ground-glass or reticular abnormalities, lung distortion, bronchiectasis or bronchioloectasis, honeycombing, and non-emphysematous cysts [20]. ILAs are considered a risk factor for developing IPF. A recent study demonstrated that ILAs have a prevalence in unselected patients of 1.7%, in 1% of the cases with fibrotic features; additionally, patients with fibrotic ILAs have a fourfold higher risk of respiratory-cause mortality, with traction bronchiectasis as significant predictor of ILA progression [21]. Thus, the correct stratification of ILAs is crucial for identifying patients at risk of progression.

Despite the introduction of MDD for the diagnosis and classification of ILDs, the agreement between MDD is moderate, considering all ILDs, particularly for non-specific interstitial pneumonia (NSIP) and hypersensitivity pneumonitis (HP) [5]. Furthermore, the course of ILD is highly variable, even for patients without a diagnosis of IPF. Some patients show a progression of ILD despite conventional therapy, a condition defined as PPF [3] The most important recent advance in ILD disease is the consistent effect of antifibrotic treatment in PPF, regardless of the etiology [4]. Nevertheless, the definition of PPF implies a follow-up period of at least 1 year, which could be critical for patient care [3]. Additionally, a recent meta-analysis stated that little evidence is available in the literature to evaluate interobserver agreement for the detection of ILD progression [22]. New insights are then required to identify patients with ILD at risk of PPF.

The interpretation of biomarkers currently employed for monitoring the ILD course can be challenging. Forced vital capacity (FVC) is considered the best index to monitor ILD; furthermore, it is used as a primary endpoint in clinical trials [4]. Nevertheless, FVC is prone to higher variability, considering that the cutoff that defines progressive disease is only a 10% decline in FVC over time [23]. A separation of the patients in a binary category based on this cutoff can be too coarse without detecting partial responders that can benefit from the antifibrotic treatment.

## 4. AI Tools in ILDs

Few algorithms have been developed for the screening of ILDs on chest X-rays or HRCT scans. The first attempt was made in 2011 by Rosas et al. for identifying ILDs in patients with familial IPF and affected by rheumatoid arthritis; the study included 126 subjects in the derivation cohort, while 86 subjects were in the validation cohort [24]. The aim of the study was to identify fibrotic and normal lung using a machine learning algorithm based on SVMs, obtaining an accuracy of 78% associated with a specificity of 70% and a sensitivity of 88% in the validation cohort [24]. More recently, another machine learning model based on SVMs, developed on 157 patients as a training set and on 40 patients as a test set, identified four features (three textural and one first-order) able to predict ILA progression at the 5-year follow-up with an area under operating curve (AUC) of 0.94 in patients enrolled for the COPD-gene group [25]. A Japanese group developed a CNN algorithm for identifying chronic interstitial lung disease on chest X-rays [26]. The tool was trained on a dataset of 653 chest X-rays with and without fibrosing ILD, giving as output a numerical score included between 0 and 1, representing the probability of fibrosis [26]. The algorithm was then tested on 120 radiographs read by expert and non-expert physicians, determining a significant increase in the performance and sensitivity in recognizing fibrotic ILD in the group of non-expert readers aided by the software with an area under the ROC curve of 0.825 vs. 0.795 (*p* = 0.005) and a sensitivity of 80% vs. 74% (*p* = 0.005), while specificity remained unchanged (84%, *p* = 0.690) [27]. Table 1 summarizes the cited main manuscript described above for the ILD screening.

More efforts have been made for the classification of ILDs with AI tools (Table 2). The first important study for the classification of fibrotic lung diseases was led by Walsh et al. in 2018 [28]. In this study, with a sample of 1157 patients (divided into 929 patients for the training set, 89 patients for the validation set, and 139 patients in test set), a deep learning algorithm was used to identify three classes of fibrotic ILDs on the basis of the ATS\ERS\JRS\ALAT guidelines provided in 2011: usual interstitial pneumonia (UIP), possible UIP, or inconsistent with UIP [29]. The algorithm showed for the classification of UIP a sensitivity of 79%, a specificity of 90%, and a C-index of 0.85; for the classification of cases inconsistent with UIP, the sensitivity was 85%, the specificity 75%, and the C-index 0.80; finally, for the classification of possible UIP, the sensitivity was 33%, the specificity 89%, and the C-index 0.61. Furthermore, the median accuracy of the 91 radiologists involved in the study was 70.7% as compared to the 73.3% accuracy of the algorithm, which outperformed humans in 66% of the cases [28]. Another important result that emerged in the study was the time required by the algorithm to classify 150 patients (four CT slices each), corresponding to around 2 s [28]. In 2019, Christe et al. used a computer-aided diagnosis (CAD) system that analyzed the HRCT of 105 patients affected by non-specific interstitial pneumonia (NSIP) and UIP, in comparison with two readers of different experience [30]. The algorithm was developed on 60 HRCT scans with annotated tissue from two radiologists and was based on a random forest analysis, which performed three steps: (1) anatomy segmentation; (2) lung tissue characterization; and (3) diagnosis. Finally, the tool differentiated the CT scans into typical UIP pattern, probable UIP pattern, indeterminate UIP pattern, and pattern more consistent with non-IPF pattern [2,30]. The algorithm showed a sensitivity of 79% and a specificity of 67% for the identification of patients that required further work-up (indeterminate UIP pattern or pattern more consistent with non-IPF pattern). The performance of the tool was similar among human readers and the AI tool, with an accuracy of around 80% for both radiologists and the algorithm [30]. Refaee et al. used an algorithm based on multiple machine and deep learning methods to differentiate IPF from non-IPF ILDs, developed on a training set of 365 patients and a test set of 109 patients, achieving a sensitivity of 83%, a specificity of 86%, and an accuracy of around 85% [31]. Another support tool for identifying the correct pattern in ILDs was proposed by Choe et al. [32]. In this study, a deep learning algorithm for the lung segmentation in ILDs was used, developed on a sample of 462, 76, and 79 HRCT scans as training, validation, and test set, respectively. Radiologists made a diagnosis of ILDs in 288 patients before and after the deep learning algorithm retrieved the three most similar images from an archive of ILDs [32]. The diagnosis accuracy significantly increased in patterns of definite UIP (81% vs. 68%, *p* < 0.001), probable UIP (67% vs. 27%, *p* < 0.001), and indeterminate UIP or alternative diagnosis (47% vs. 17%, *p* = 0.003) [32].

Several studies investigated computer-aided tools for prognostication of ILDs. One of the most promising software toolswas computer-aided lung informatics for pathology evaluation and rating (CALIPER), based on a machine learning approach [33]. CALIPER is able to quantify automatically at HRCT the ILD extent, fibrosis extent, honeycombing extent, traction bronchiectasis severity, traction bronchiectasis extent, and vessel-related structures [33]. The CALIPER quantification showed a better correlation with PFTs as compared to the visual score; furthermore, CALIPER-derived features, particularly vessel-related structures (corresponding to pulmonary vessel volume added to perivascular fibrosis), are strong predictors of 12-month mortality or FVC decline [33,34]. Another adaptive model software was trained on 307 HRCTs in which three expert radiologists labeled ground glass, ground-glass reticulations, bronchovascular, emphysema, normal lung, and honeycombing in patients affected by IPF [35]. In the multivariable analysis, the quantification of the sum of reticulations and ground glass performed by the software was an independent, significant, predictor of disease progression, defined as a composite outcome of FVC decline>10%, hospitalization, or mortality (HR 1.36, *p* = 0.04) [35]. Xu et al. developed an algorithm based on radiomic features to predict 6-month all-cause mortality in baseline HRCT scans of patients affected by anti-melanoma differentiation-associated gene 5-positive dermatomyositis-associated interstitial lung disease (MDA5+ DM-ILD); the algorithm was developed on a training set of 121 patients and on a testing set of 31 patients, with an internal validation cohort of 44 patients and an external validation cohort of 32 patients [36]. The predictive model, including selected radiomics features, age, and functional data, outperformed the HRCT visual assessment at baseline (C-index 0.88 vs. 0.75) [36]. Recently, Walsh et al. used a novel approach for predicting at 12 months functional decline (>10% FVC decline or >15% DLCO reduction), death, or transplant [37]. With the aid of a deep learning method (systematic objective fibrotic imaging analysis algorithm, SOFIA) developed on a training set of 1157 HRCTs (four slice montage) and on a test set of 150 HRCTs, UIP probabilities in a cohort (n = 504) of progressive fibrotic diseases were converted from prospective investigation of pulmonary embolism (PIOPED)-based UIP probability categories (UIP not included in the differential, 0–4%; low-probability of UIP, 5–29%; intermediate probability of UIP, 30–69%; high probability of UIP, 70–94%; and pathognomonic for UIP, 95–100%) [37]. SOFIA-PIOPED UIP probability was a significant predictor for ILD worse outcome (HR = 1.29, *p* < 0.0001), even in patients considered indeterminate for UIP at visual assessment by radiologists; this tool improved the outcome prediction as compared to expert radiologists’ evaluation, and it could be particularly useful in multidisciplinary discussions of challenging cases, particularly those indeterminate for UIP pattern [37]. A more comprehensive approach recently developed is based on the use of different deep learning algorithms that analyze HRCT scans with clinical and functional data at baseline and during follow-up; the algorithm was developed on a training set of 258 cases, a validation set of 63 cases, and a test set of 128 patients [38]. First the algorithm categorizes ILDs as chronic hypersensitivity pneumonitis (AUC, 0.814), NSIP (AUC, 0.849), sarcoidosis (AUC, 0.740), UIP (AUC, 0.828), or other ILD (AUC, 0.788), then provides an individualized probability of survival at 3 years (Figure 7) [38]. The model showed better performance than humans for the diagnosis of UIP (AUC, 0.828 vs. 0.763; sensitivity, 82% vs. 55%, *p*<0.05; specificity, 68% vs. 97%, *p* < 0.001) and showed an AUC of 0.868 with a sensitivity of 73% and a specificity of 84% for the prediction of 3-year survival [38].

In relation to the recent developments in the treatment of progressive fibrotic ILDs, a variety of different softwares has been tested to aid radiologists for detecting at HRCT the increasing extent of pulmonary fibrosis. In one series, 468 patients with ILD were selected and stratified into three groups based on FVC decline over 24 months (less than 5%, included between 5% and 9.9%, and greater than or equal to 10%) [39]. Commercially available software based on deep learning texture analysis, quantified both at baseline and follow-up (median follow-up 44 months), HRCT ground-glass opacities, consolidation, reticular opacity, and honeycombing; the sum of reticulations, ground glass, and honeycombing was considered ILD extent, while reticulations plus honeycombing was considered fibrosis [39]. The inclusion of computer-aided quantification of both increasing ILD extent (*p* = 0.01) and increasing fibrosis (*p* = 0.02) associated with age, sex, initial %FVC, and decline of FVC in 24 months higher than 10% significantly improved the model performance for predicting all-cause mortality, while visual assessment by radiologists failed to increment model performance (*p* = 0.99) [39]. Table 3 shows AI studies regarding prognosis and follow-up of ILDs.

## 5. Discussion

Computer-aided evaluation demonstrated, in several studies available in the literature during recent years, to have the opportunity and the potentiality to overcome the screening, diagnostic, prognostic, and monitoring challenges of ILDs.

The prognosis of established ILDs and stratification of ILAs represents the same issue but in different time-points of the disease (early in ILAs while in more advanced stages for established ILDs). A reliable stratification of ILAs visually is a difficult task considering the challenging quantification of small areas of the lung and the high interobserver variability [40]. Predicting disease behavior at the baseline of established ILDs is similarly difficult. Currently it is well known that the UIP-pattern can progress over time and the extent of fibrotic changes at baseline is a predictor of worse outcome, likely because in the past a progression has occurred. Deep learning has the potentiality to recognize patterns, even difficult ones, classifying them objectively and giving a probability of disease progression or mortality. Furthermore, it is possible to train algorithms not on labeled images (subjectively by radiologists) but by anchoring them to objective data, such as FVC decline, overcoming the physiological inter-reader human variability. As previously mentioned, SOFIA showed enthusiastic results for the prognosis of IPF in a huge national registry [37].

Nevertheless, the development of AI in ILDs has several limitations. First, ILDs are rare diseases divided into a huge number of different diseases with not completely predictable behavior. Therefore, collaboration in data collecting is of paramount importance. Clinical and radiological data collection should be standardized at specific time-points. Furthermore, standardized protocols of acquisition of HRCT between different manufacturers are required to analyze more homogeneous images, and clinical data should be centralized in international registries; with regard to this issue, data governance should reduce restrictions for the access of clinical and radiological information. Second, algorithm transparency in terms of training and validation is a key assumption for approval by regulatory bodies for the implementation in daily clinical routine. Third, the validation of computer-aided software in clinical practice is crucial for the usefulness of these algorithms. New computer-aided biomarkers should be tested against currently used staging systems (e.g., GAP stage), with prospective trials, and should be compared to each other to estimate their reliability. 

## 6. Conclusions

In conclusion, in recent years it has been observed the development of a huge number of software and algorithms applied in ILDs, that showed similar or better performances as compared to humans for screening, diagnosis, prognosis, and monitoring of ILDs in the research field. In particular, the approach that includes deep-learning analysis of radiological and clinical data during different time-points is the most intriguing to accomplish the aim of precision and individualized treatment. Nevertheless, for translating in clinical practice the innovation demonstrated in the literature by AI tools, more efforts are required to collect more homogeneous radiological and clinical data in international registries and to validate the reliability of these algorithms.

## Figures and Tables

**Figure 1 diagnostics-15-00943-f001:**
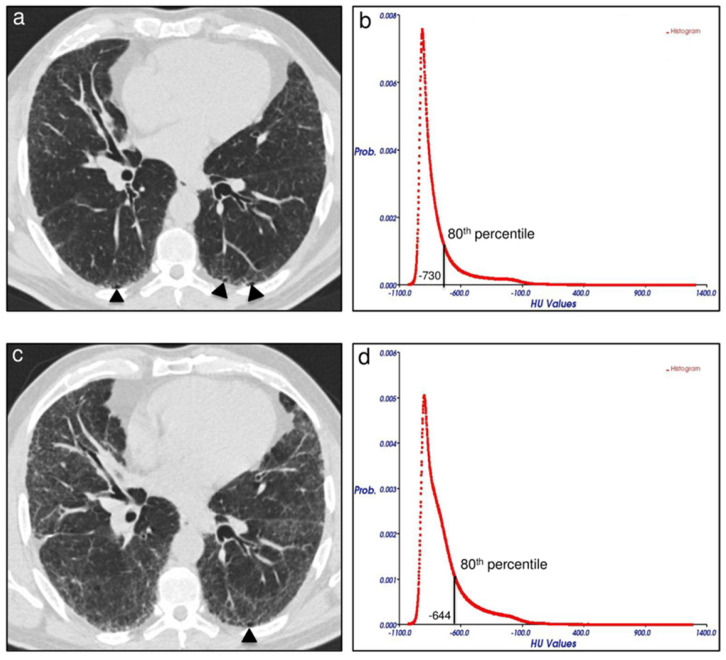
An example of quantitative computed tomography (QCT) applied in interstitial lung disease; in idiopathic pulmonary fibrosis (IPF) the increase over time (17 months) in density (HU) of the 80th percentile of the lung density histogram (**d**) as compared to baseline (**b**), reflected reticulation progression as depicted visually in high-resolution computed tomography (HRCT) images at follow-up (**c**) in comparison to baseline HRCT (**a**) (adapted by Colombi et al.) [13].

**Figure 2 diagnostics-15-00943-f002:**
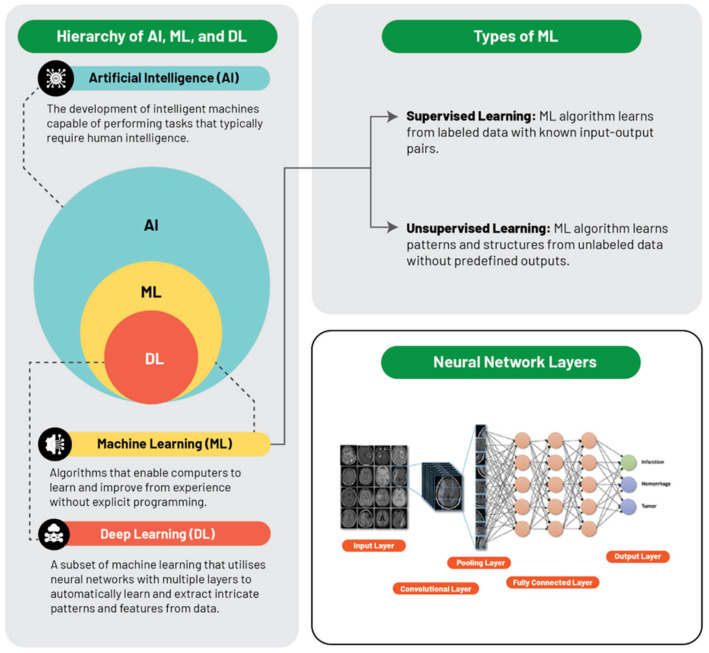
Hierarchy of the artificial intelligence tool in medical images (adapted by Najjar et al.) [16].

**Figure 3 diagnostics-15-00943-f003:**
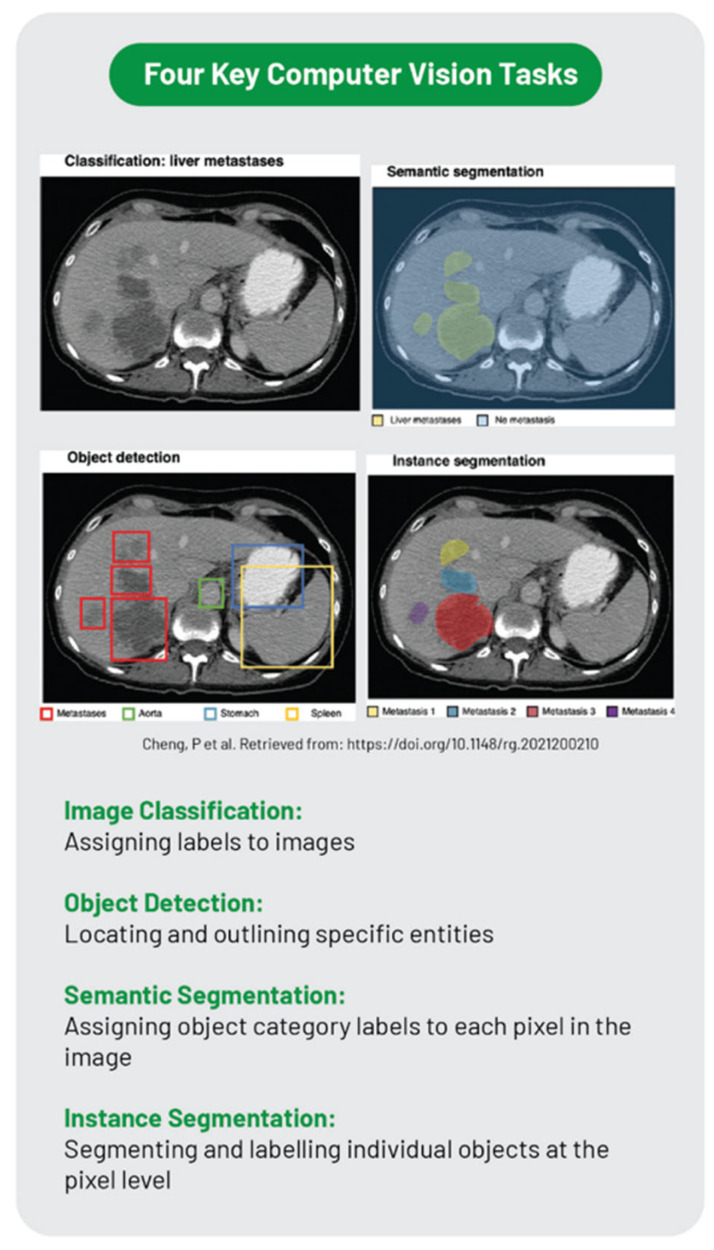
Computer vision tasks in deep learning (adapted by Najjar et al.) [16,17].

**Figure 4 diagnostics-15-00943-f004:**
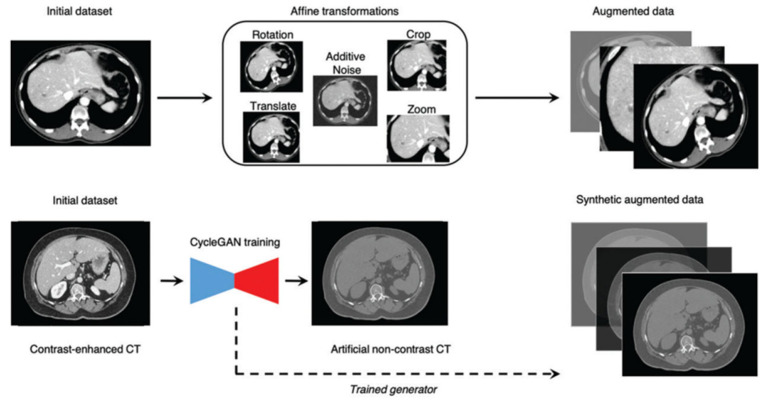
Data augmentation based on affine transformations or generative adversarial networks (GANs) (adapted by Cheng et al.) [17].

**Figure 5 diagnostics-15-00943-f005:**
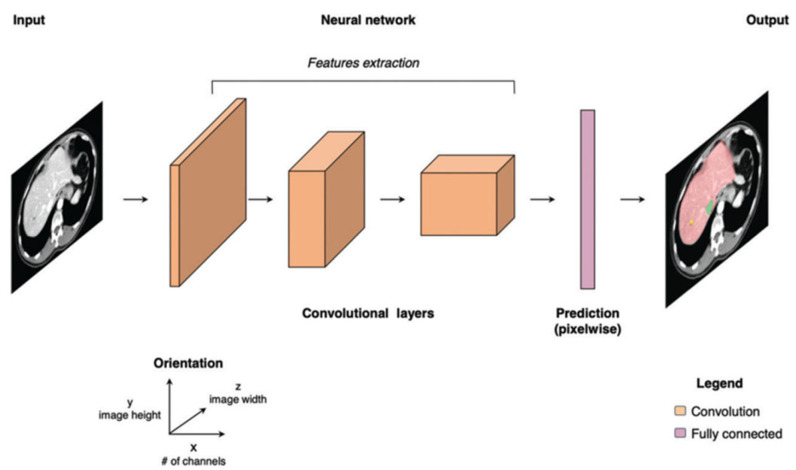
Convolutional neural network diagram with input data that finally reach an output (in the case presented, recognition of lesions and organs) after convolutional and pooling serial layers (adapted by Cheng et al.) [17].

**Figure 6 diagnostics-15-00943-f006:**
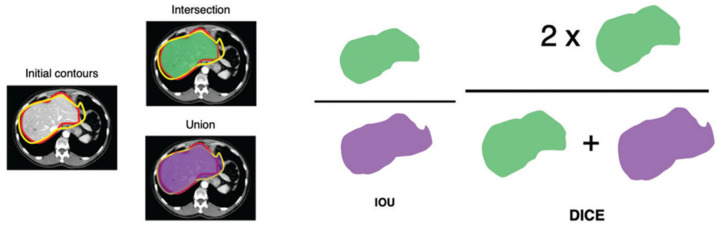
Intersection over union (IOU) and DICE are methods that evaluate the degree of overlap in segmentation of an object as compared to human segmentation (adapted by Cheng et al.) [17].

**Figure 7 diagnostics-15-00943-f007:**
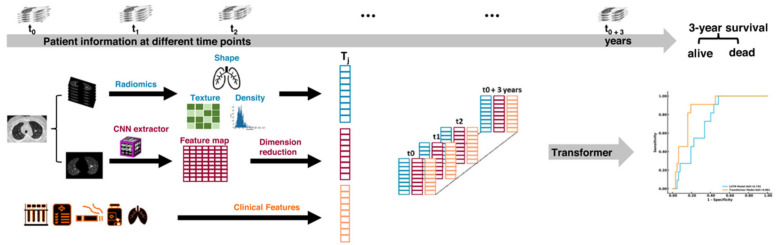
The illustration shows the AI model developed by Mei et al. that included high-resolution computed tomography (HRCT) and clinical data at different time-points to predict survival probability at three years (adapted by Mei et al.) [38].

**Table 1 diagnostics-15-00943-t001:** Summary of the AI studies for the screening of ILDs.

Study (Year)	Population	Algorithm Task
Rosas et al. (2011) [24]	Patients with familial fibrosis or rheumatoid arthritis with or without ILDs	Identify at HRCT normal findings or ILDs
Poynton et al. (2021) [25]	Patients with ILAs and 5-year follow-up	Identify at baseline HRCT features of ILA progression
Ukita et al. (2024) [27]	Patients with or without ILDs at chest X-ray	Aid radiologist to identify ILDs at chest X-ray

Abbreviations: AI, artificial intelligence; HRCT, high-resolution computed tomography; ILAs, interstitial lung abnormalities; ILDs, interstitial lung diseases.

**Table 2 diagnostics-15-00943-t002:** Summary of the AI studies for the classification of ILDs.

Study (Year)	Population	Algorithm Task
Walsh et al. (2018) [28]	Patients with pulmonary fibrosis	Classification of HRCT in UIP, possible UIP, or inconsistent with UIP pattern [29]
Christe et al. (2019) [30]	Patients with pulmonary fibrosis	Classification of HRCT in typical UIP, probable UIP, indeterminate UIP, or non-IPF pattern [2]
Refaee et al. (2022) [31]	Patients with ILDs	Classification of HRCT in IPF vs. non-IPF ILDs
Choe et al. (2022) [32]	Patients with ILDs	Proposed content-based image retrieval to aid radiologist classification of ILDs at HRCT

Abbreviations: AI, artificial intelligence; HRCT, high-resolution computed tomography; ILDs, interstitial lung diseases; IPF, idiopathic pulmonary fibrosis; UIP, usual interstitial pneumonia.

**Table 3 diagnostics-15-00943-t003:** Summary of the AI studies for the prognosis and follow-up of ILDs.

Study (Year)	Population	Algorithm Task
Salisbury et al. (2017) [35]	Patients with IPF	Prediction of mortality, hospitalization, or FVC decline by software fibrosis quantification at baseline HRCT
Jacob et al. (2018) [34]	Patients with IPF	Prediction of mortality or FVC decline by computer-derived features at baseline HRCT with machine learning algorithm (CALIPER)
Xu et al. (2021) [36]	Patients with MDA5+ DM-ILD	Prediction of mortality by software features obtained at baseline HRCT
Walsh et al. (2022) [37]	Patients with progressive fibrotic lung disease	Prediction of mortality by PIOPED-derived classification of UIP at baseline HRCT obtained by a deep learning algorithm (SOFIA)
Mei et al. (2023) [38]	Patients with ILDs	Prediction of mortality by a deep learning algorithm developed on HRCT, clinical, and functional data obtained over time
Koh et al. (2024) [39]	Patients with ILDs	Prediction of mortality by quantification of ILD extent and ILD fibrosis increase over time quantified by fully automatic software

Abbreviations: CALIPER, computer-aided lung informatics for pathology evaluation and rating; MDA5+ DM-ILD, anti-melanoma differentiation-associated gene 5-positive dermatomyositis-associated interstitial lung disease; FVC, forced vital capacity; HRCT, high-resolution computed tomography; ILDs, interstitial lung diseases; IPF, idiopathic pulmonary fibrosis; PIOPED, prospective investigation of pulmonary embolism diagnosis; SOFIA, systematic objective fibrotic imaging analysis algorithm; UIP, usual interstitial pneumonia.

## Data Availability

No new data were created or analyzed in this study. Data sharing is not applicable to this article.

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
