# Peer review of "Computer-Aided Evaluation of Interstitial Lung Diseases"

_diagnostics, 2025, doi:10.3390/diagnostics15070943_

Round 1
Reviewer 1 Report
Comments and Suggestions for Authors
Excellent review on current status of AI tools available to diagnose, characterize and prognosticate ILD's.
Brief summary of AI, Machine learning and Deep learning for non AI professionals is appropriate to understand the basics of the technical details. Brief summary of the available studies has been put in context of the potential and shortcomings in the field so far.
The conclusion aptly describes the challenges in the field due to the incredible diversity of causes, radiological presentations and natural course of the disease.
Highly recommend the article to radiologists and pulmonologists involved in ILD and specifically research into these rare but deadly diseases.
Author Response
Comment 1: Excellent review on current status of AI tools available to diagnose, characterize and prognosticate ILD's.
Brief summary of AI, Machine learning and Deep learning for non AI professionals is appropriate to understand the basics of the technical details. Brief summary of the available studies has been put in context of the potential and shortcomings in the field so far.
The conclusion aptly describes the challenges in the field due to the incredible diversity of causes, radiological presentations and natural course of the disease.
Highly recommend the article to radiologists and pulmonologists involved in ILD and specifically research into these rare but deadly diseases.
Response 1: thank you, we are delighted about your insightful comments.
Reviewer 2 Report
Comments and Suggestions for Authors
The review is generally well written. I have only a few minor comments:
- The authors write: "Typically, MR images have a resolution of 256x256 while CT images of 512x512". 256x256 is the size of the image, in DICOM header there is also pixel size. Resolution, according to a common definition, refers to the minimal distance of some two distinct objects such that the objects can be perceived as separated.
- The authors wtite "The most common unsupervised models are neural networks." Neural networks can be used for unsupervised tasks (e.g. autoencoders) but the most common medical applications of neural networks are supervised tasks: segmentation, classification, object detection.
- The authors write "Neural networks work as the human brain, with input layers, several middle layers, and an output layer." which is a common misunderstanding. Neural networks are just nonlinear functions which on tensors. I suggest removing the text from "Neural networks work as the human brain" to "The analysis continues until an output layer".
- The authors write "The first layer is a convolution layer in which a 3x3 pixels kernel (which is a 2D matrix) is..." The details of a deep model depend on application so different deep models can have different kernels, may or may not use pooling, can be 2D or 3d and so on.
- The authors write "The error is defined as the sum of 1 minus the correct output node and plus all the other nodes. The error calculated is then used as a weight of the CNN algorithm. " The loss function depends on application and is different for classification, segmentation, object detection. The gradient of loss is used for modifying weights.
- In AI tools in ILDs section the authors should provide more details about the studies (if available) like how big were the datasets used, how these datasets were split into training/validation/test parts; accuracy/sensitivity/specificity to make the comparison more comprehensive
Author Response
Comment 1: The review is generally well written. I have only a few minor comments:
Response 1: thank you, we are delighted about you insightful comments.
Comment 2: The authors write: "Typically, MR images have a resolution of 256x256 while CT images of 512x512". 256x256 is the size of the image, in DICOM header there is also pixel size. Resolution, according to a common definition, refers to the minimal distance of some two distinct objects such that the objects can be perceived as separated.
Response: 2: thank you for this important comment, we have amended it in the text (Page 2, line 54).
Comment 3: The authors write "The most common unsupervised models are neural networks." Neural networks can be used for unsupervised tasks (e.g. autoencoders) but the most common medical applications of neural networks are supervised tasks: segmentation, classification, object detection.
Response 3: thank you for your suggestion. We have changed the text as indicated (Page 4, lines 162-163).
Comment 4: The authors write "Neural networks work as the human brain, with input layers, several middle layers, and an output layer." which is a common misunderstanding. Neural networks are just nonlinear functions which on tensors. I suggest removing the text from "Neural networks work as the human brain" to "The analysis continues until an output layer".
Response 4: thank you for your insightful comment. We have modified the sentence as suggested (Page 4, line 164).
Comment 5: The authors write "The first layer is a convolution layer in which a 3x3 pixels kernel (which is a 2D matrix) is..." The details of a deep model depend on application so different deep models can have different kernels, may or may not use pooling, can be 2D or 3d and so on.
Response 5: thank you for this main explanation. We have amended the text as suggested (Pages 5-6, lines 200-204).
Comment 6: The authors write "The error is defined as the sum of 1 minus the correct output node and plus all the other nodes. The error calculated is then used as a weight of the CNN algorithm. " The loss function depends on application and is different for classification, segmentation, object detection. The gradient of loss is used for modifying weights.
Response 6: thank you for this main remark. We have amended the text as suggested (Page 6, lines 211-213).
Comment 7: In AI tools in ILDs section the authors should provide more details about the studies (if available) like how big were the datasets used, how these datasets were split into training/validation/test parts; accuracy/sensitivity/specificity to make the comparison more comprehensive.
Response 7: thank you for this important remark. We have specified the data requested (when available) in the section (Pages 8-9-10-11, lines 292-418).